# No Pressure! Addressing the Problem of Local Minima in Manifold Learning Algorithms

**Max Vladymyrov**
Google Research
mxv@google.com

## Abstract

Nonlinear embedding manifold learning methods provide invaluable visual insights into the structure of high-dimensional data. However, due to a complicated nonconvex objective function, these methods can easily get stuck in local minima and their embedding quality can be poor. We propose a natural extension to several manifold learning methods aimed at identifying pressured points, i.e. points stuck in poor local minima and have poor embedding quality. We show that the objective function can be decreased by temporarily allowing these points to make use of an extra dimension in the embedding space. Our method is able to improve the objective function value of existing methods even after they get stuck in a poor local minimum.

## 1 Introduction

Given a dataset $\mathbf{Y} \in \mathbb{R}^{D \times N}$ of $N$ points in some high-dimensional space with dimensionality $D$, manifold learning algorithms try to find a low-dimensional embedding $\mathbf{X} \in \mathbb{R}^{d \times N}$ of every point from $\mathbf{Y}$ in some space with dimensionality $d \ll D$. These algorithms play an important role in high-dimensional data analysis, specifically for data visualization, where $d = 2$ or $d = 3$. The quality of the methods have come a long way in recent decades, from classic linear methods (e.g. PCA, MDS), to more nonlinear spectral methods, such as Laplacian Eigenmaps [Belkin and Niyogi, 2003], LLE [Saul and Roweis, 2003] and Isomap [de Silva and Tenenbaum, 2003], finally followed by even more general nonlinear embedding (NLE) methods, which include Stochastic Neighbor Embedding (SNE, Hinton and Roweis, 2003), $t$-SNE [van der Maaten and Hinton, 2008], NeRV [Venna et al., 2010] and Elastic Embedding (EE, Carreira-Perpiñán, 2010). This last group of methods is considered as state-of-the-art in manifold learning and became a go-to tool for high-dimensional data analysis in many domains (e.g. to compare the learning states in Deep Reinforcement Learning [Mnih et al., 2015] or to visualize learned vectors of an embedding model [Kiros et al., 2015]).

While the results of NLE have improved in quality, their algorithmic complexity has increased as well. NLE methods are defined using a nonconvex objective that requires careful iterative minimization. A lot of effort has been spent on improving the convergence of NLE methods, including Spectral Direction [Vladymyrov and Carreira-Perpiñán, 2012] that uses partial-Hessian information in order to define a better search direction, or optimization using a Majorization-Minimization approach [Yang et al., 2015]. However, even with these sophisticated custom algorithms, it is still often necessary to perform a few random restarts in order to achieve a decent solution. Sometimes it is not even clear whether the learned embedding represents the structure of the input data, noise, or the artifacts of an embedding algorithm [Wattenberg et al., 2016].

Consider the situation in fig. 1. There we run the EE 100 times on the same dataset with the same parameters, varying only the initialization. The dataset, COIL-20, consists of photos of 20 different objects as they are rotated on a platform with new photo taken every 5 degrees (72 images per object). Good embedding should separate objects one from another and also reflect the rotational sequence of each object (ideally via a circular embedding). We see in the left plot that for virtually every run the embedding gets stuck in a distinct local minima. The other two figures show the difference between the best and the worst embedding depending on how lucky we get with the initialization.

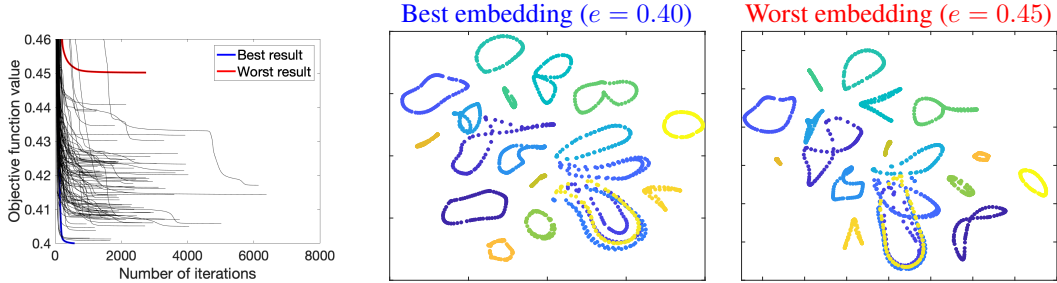

Figure 1: Abundance of local minima in the Elastic Embedding objective function space. We run the algorithm $100\times$ on COIL-20 dataset with different random initializations. We show the objective function decrease (*left*), the embedding result for the run with the lowest (*center*) and the highest (*right*) final objective function values. Color encodes different objects.

The embedding in the center has much better quality compared to the one on the right, since most of the objects are separated from each other and their embeddings more closely resemble a circle.

In this paper we focus on the analysis of the reasoning behind the occurrence of local minima in the NLE objective function and ways for the algorithms to avoid them. Specifically, we discuss the conditions under which some points get caught in high-energy states of the objective function. We call these points "pressured points" and show that specifically for the NLE class of algorithms there is a natural way to identify and characterize them during optimization.

Our contribution is twofold. First, we look at the objective function of the NLE methods and provide a mechanism to identify the pressured points for a given embedding. This can be used on its own as a diagnostic tool for assessing the quality of a given embedding at the level of individual points. Second, we propose an optimization algorithm that is able to utilize the insights from the pressured points analysis to achieve better objective function values even from a converged solution of an existing state-of-the-art optimizer. The proposed modification augments the existing analysis of the NLE and can be run on top of state-of-the-art optimization methods: Spectral Direction and $N$-body algorithms [Yang et al., 2013, van der Maaten, 2014, Vladymyrov and Carreira-Perpiñán, 2014].

Our analysis arises naturally from a given NLE objective function and does not depend on any other assumptions. Other papers have looked into the problem of assessing the quality of the embedding [Peltonen and Lin, 2015, Lee and Verleysen, 2009, Lespinats and Aupetit, 2011]. However, their quality criteria are defined separately from the actual learned objective function, which introduces additional assumptions and does not connect to the original objective function. Moreover, we also propose a method for improving the embedding quality in addition to assessing it.

## 2 Nonlinear Manifold Learning Algorithms

The objective functions for SNE and $t$-SNE were originally defined as a KL-divergence between two normalized probability distributions of points being in the neighborhood of each other. They use a positive affinity matrix $\mathbf{W}^+$, usually computed as $w_{ij}^+ = \exp(-\frac{1}{2\sigma^2}\|\mathbf{y}_i - \mathbf{y}_j\|^2)$, to capture a similarity of points in the original space $D$. The algorithms differ in the kernels they use in the low-dimensional space. SNE uses the normalized Gaussian kernel[1] $K_{ij} = \frac{\exp(-\|\mathbf{x}_i-\mathbf{x}_j\|^2)}{\sum_{n,m}\exp(-\|\mathbf{x}_n-\mathbf{x}_m\|^2)}$, while $t$-SNE is using the normalized Student's $t$ kernel $K_{ij} = \frac{(1+\|\mathbf{x}_i-\mathbf{x}_j\|^2)^{-1}}{\sum_{n,m}(1+\|\mathbf{x}_n-\mathbf{x}_m\|^2)^{-1}}$.

UMAP [McInnes et al., 2018] uses the unnormalized kernel $K_{ij} = \left(1 + a\,\|\mathbf{x}_i - \mathbf{x}_j\|^{2b}\right)^{-1}$ that is similar to Student's $t$, but with additional constants $a, b$ calculated based on the topology of the original manifold. The objective function is given by the cross entropy as opposed to KL-divergence.

Carreira-Perpiñán [2010] showed that these algorithms could be defined as an interplay between two additive terms: $E(\mathbf{X}) = E^+(\mathbf{X}) + E^-(\mathbf{X})$. Attractive term $E^+$, usually convex, pulls points close to each other with a force that is larger for points located nearby in the original space. Repulsive term $E^-$, on the contrary, pushes points away from each other. For SNE and $t$-SNE the attraction is given by the nominator of the normalized kernel, while the repulsion is the denominator. It intuitively makes sense, since in order to pull some point closer (decrease the nominator), you have to push all the other points away a little bit (increase the denominator) so that the probability would still sum

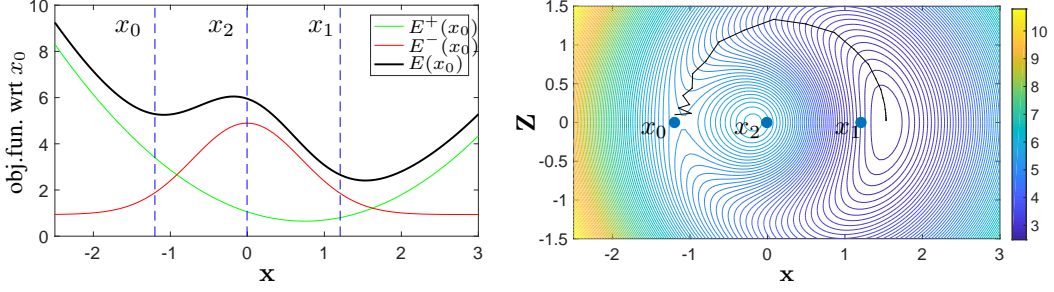

Figure 2: *Left:* an illustration of the local minimum typically occurring in NLE optimization. Blue dashed lines indicate the location of 3 points in 1D. The curves show the objective function landscape wrt $x_0$. *Right:* by enabling an extra dimension for $x_0$, we can create a "tunnel" that avoids a local minimum in the original space, but follows a continuous minimization path in the augmented space.

to one. For UMAP, there is no normalization to act as a repulsion, but the repulsion is given by the second term in the cross entropy (i.e. the entropy of the low-dimensional probabilities).

Elastic Embedding (EE) modifies the repulsive term of the SNE objective by dropping the $\log$, adding a weight $\mathbf{W}^-$ to better capture non-local interactions (e.g. as $w_{ij}^- = \|\mathbf{y}_i - \mathbf{y}_j\|^2$), and introducing a scaling hyperparameter $\lambda$ to control the interplay between two terms.

Here are the objective functions of the described methods:

$$E_{\text{EE}}(\mathbf{X}) = \sum_{i,j} w_{ij}^+ \|\mathbf{x}_i - \mathbf{x}_j\|^2 + \lambda \sum_{i,j} w_{ij}^- e^{-\|\mathbf{x}_i - \mathbf{x}_j\|^2}, \tag{1}$$

$$E_{\text{SNE}}(\mathbf{X}) = \sum_{i,j} w_{ij}^+ \|\mathbf{x}_i - \mathbf{x}_j\|^2 + \log \sum_{i,j=1} e^{-\|\mathbf{x}_i - \mathbf{x}_j\|^2}, \tag{2}$$

$$E_{t\text{-SNE}}(\mathbf{X}) = \sum_{i,j} w_{ij}^+ \log \left(1 + \|\mathbf{x}_i - \mathbf{x}_j\|^2\right) + \log \sum_{i,j} \frac{1}{1 + \|\mathbf{x}_i - \mathbf{x}_j\|^2}, \tag{3}$$

$$E_{\text{UMAP}}(\mathbf{X}) = \sum_{i,j} w_{ij}^+ \log \left(1 + a \|\mathbf{x}_i - \mathbf{x}_j\|^{2b}\right) + \sum_{i,j} (w_{ij}^+ - 1) \log(1 - \frac{1}{1 + a\|\mathbf{x}_i - \mathbf{x}_j\|^{2b}}). \tag{4}$$

## 3 Identifying pressured points

Let us consider the optimization with respect to a given point $\mathbf{x}_0$ from $\mathbf{X}$. For all the algorithms the attractive term $E^+$ grows as $\|\mathbf{x}_0 - \mathbf{x}_n\|^2$ and thus has a high penalty for points placed far away in the embedding space (especially if they are located nearby in the original space). The repulsive term $E^-$ is mostly localized and concentrated around individual neighbors of $\mathbf{x}_0$. As $\mathbf{x}_0$ navigates the landscape of $E$ it tries to get to the minimum of $E^+$ while avoiding the "hills" of $E^-$ created around repulsive neighbors. However, the degrees of freedom of $\mathbf{X}$ is limited by $d$ which is typically much smaller than the intrinsic dimensionality of the data. It might happen that the point gets stuck surrounded by its non-local neighbors and is unable to find a path through.

We can illustrate this with a simple scenario involving three points $\mathbf{y}_0, \mathbf{y}_1, \mathbf{y}_2$ in the original $\mathbb{R}^D$ space, where $\mathbf{y}_0$ and $\mathbf{y}_1$ are near each other and $\mathbf{y}_2$ is further away. We decrease the dimensionality to $d = 1$ using EE algorithm and assume that due to e.g. poor initialization $x_2$ is located between $x_0$ and $x_1$. In the left plot of fig. 2 we show different parts of the objective function as a function of $x_0$. The attractive term $E^+(x_0)$ creates a high pressure for $x_0$ to move towards $x_1$. However, the repulsion between $x_0$ and $x_2$ creates a counter pressure that pushes $x_0$ away from $x_2$, thus creating two minima: one local near $x = -1$ and another global near $x = 1.5$. Points like $x_0$ are trapped in high energy regions and are not able to move. We argue that these situations are the reason behind many of the local minima of NLE objective functions. By identifying and repositioning these points we can improve the objective function and overall the quality of the embedding.

We propose to evaluate the pressure of every point with a very simple and intuitive idea: increased pressure from the "false" neighbors would create a higher energy for the point to escape that location. However, for a true local minimum, there are no directions for that point to move. That is, given the *existing* number of dimensions. If we were to add a new dimension $\mathbf{Z}$ temporarily just for that point, it would be possible for the points to move along that new dimension (see fig. 2, right). The more that point is pressured by other points, the farther across this new dimension it would go.

More formally, we say that the point is *pressured* if the objective function has a nontrivial minimum when evaluated at that point along the new dimension $\mathbf{Z}$. We define the minimum $\hat{z}$ along the dimension $\mathbf{Z}$ as the pressure of that point.

It is important to notice the distinction between pressured points and points that have higher objective function value when evaluated at those points (a criterion that is used e.g. in Lespinats and Aupetit [2011] to assess the embedding quality). Large objective function value alone does not necessarily mean that the point is stuck in a local minimum. First, the point could still be on its way to the minimum. Second, even for an embedding that represents the global minimum, each point would converge to its own unique objective function value since the affinities for every point are distinct. Finally, not every NLE objective function can be easily evaluated for every point separately. SNE (2) and $t$-SNE (4) objective functions contain $\log$ term that does not allow for easy decoupling.

In what follows we are going to characterize the pressure of each point and look at how the objective function changes when we add an extra dimension to each of the algorithms described above.

**Elastic Embedding.** For a given point $k$ we extend the objective function of EE (1) along the new dimension $\mathbf{Z}$. Notice that we consider points individually one by one, therefore all $z_i = 0$ for all $i \neq k$. The objective function of EE along the new dimension $z_k$ becomes:

$$\widetilde{E}_{\text{EE}}(z_k) = 2z_k^2 d_k^+ + 2\tilde{d}_k^- e^{-z_k^2} + C, \tag{5}$$

where $d_k^+ = \sum_{i=1}^N w_{ik}^+$, $\tilde{d}_k^- = \lambda \sum_{i=1}^N w_{ik}^- e^{-\|\mathbf{x}_i - \mathbf{x}_k\|^2}$ and $C$ is a constant independent from $z_k$. The function is symmetric wrt 0 and convex for $z_k \geq 0$. Its derivative is

$$\frac{\partial \widetilde{E}_{\text{EE}}(z_k)}{\partial z_k} = 4z_k \left( d_k^+ - e^{-z_k^2} \tilde{d}_k^- \right). \tag{6}$$

The function has a stationary point at $z_k = 0$, which is a minimum when $\tilde{d}_k^- < d_k^+$. Otherwise, $z_k = 0$ is a maximum and the only non-trivial minimum is $\hat{z}_k = \sqrt{\log(\tilde{d}_k^- / d_k^+)}$. The magnitude of the fraction under the log corresponds to the amount of pressure for $\mathbf{x}_k$. The numerator $\tilde{d}_k^-$ depends on $\mathbf{X}$ and represents the pressure that the neighbors of $\mathbf{x}_k$ exert on it. The denominator is given by the diagonal element $k$ of the degree matrix $\mathbf{D}^+$ and represents the attraction of the points in the original high-dimensional space. The fraction is smallest when points are ordered by $w_{ik}^-$ for all $i \neq k$, i.e. ordered by distance from $\mathbf{y}_k$. As points change order and move closer to $\mathbf{x}_k$ (especially those far in the original space, i.e. with high $w_{ik}^-$) $\tilde{d}_k^-$ increases and eventually turns $\widetilde{E}_{\text{EE}}(z_k = 0)$ from a minimum to a maximum, thus creating a pressured point.

**Stochastic Neighbor Embedding.** The objective along the dimension $\mathbf{Z}$ for a point $k$ is given by:

$$\widetilde{E}_{\text{SNE}}(z_k) = 2z_k^2 d_k^+ + \log \left( 2(e^{-z_k^2} - 1)\tilde{d}_k^- + \sum_n \tilde{d}_n^- \right) + C,$$

where, slightly abusing the notation between different methods, we define $d_k^+ = \sum_{i=1}^N w_{ik}^+$ and $\tilde{d}_k^- = \sum_{i=1}^N e^{-\|\mathbf{x}_i - \mathbf{x}_k\|^2}$. The derivative is equal to

$$\frac{\partial \widetilde{E}_{\text{SNE}}(z_k)}{\partial z_k} = 4z_k \left( d_k^+ - \frac{e^{-z_k^2} \tilde{d}_k^-}{2(e^{-z_k^2}-1)\tilde{d}_k^- + \sum_n \tilde{d}_n^-} \right).$$

Similarly to EE, the function is convex, has a stationary point at $z_k = 0$, which is a minimum when $\tilde{d}_k^-(1 - 2d_k^+) < d_k^+ \left( \sum_n \tilde{d}_n^- - 2\tilde{d}_k^- \right)$. It also can be rewritten as $\frac{\sum_{i=1}^N \exp(-\|\mathbf{x}_i - \mathbf{x}_k\|^2)}{\sum_{i,j \neq k}^N \exp(-\|\mathbf{x}_i - \mathbf{x}_j\|^2)} < \frac{\sum_{i=1}^N w_{ik}^+}{\sum_{i,j \neq k}^N w_{ij}^+}$. The LHS represents the pressure of the points on $\mathbf{x}_k$ normalized by an overall pressure for the rest of the points. If this pressure gets larger than the similar quantity in the original space (RHS), the point becomes pressured with the minimum at $\hat{z}_k = \sqrt{\log \frac{\tilde{d}_k^-(1-2d_k^+)}{d_k^+ \left( \sum_n \tilde{d}_n^- - 2\tilde{d}_k^- \right)}}$.

**$t$-SNE.** $t$-SNE uses Student's $t$ distribution which does not decouple as nice as the Gaussian kernel for EE and SNE. The objective along $z_k$ and its derivative are given by

$$\widetilde{E}_{t\text{-SNE}}(z_k) = 2 \sum_{i=1}^N w_{ik} \log \left( K_{ik}^{-1} + z_k^2 \right) + \log \left( \sum_{i,j \neq k}^N K_{ij} + \sum_{i=1}^N \frac{2}{K_{ik}^{-1} + z_k^2} \right) + C.$$

$$\frac{\partial \widetilde{E}_{t\text{-SNE}}(z_k)}{\partial z_k} = 4z_k \left( \sum_{i=1}^N \frac{w_{ik}^+}{K_{ik}^{-1} + z_k^2} - \frac{\sum_{i=1}^N (K_{ik}^{-1} + z_k^2)^{-2}}{\sum_{i,j \neq k}^N K_{ij} + 2\sum_{i=1}^N (K_{ik}^{-1} + z_k^2)^{-1}} \right).$$

where $K_{ij} = (1 + \|\mathbf{x}_i - \mathbf{x}_j\|^2)^{-1}$. The function is convex, but the closed form solution is now harder to obtain. Practically it can be done with just a few iterations of the Newton's method initialized at some positive value close to 0. In addition, we can quickly test whether the point is pressured or not from the sign of the second derivative at $z_k = 0$: $\frac{\partial \widetilde{E}_{t\text{-SNE}}^2(0)}{\partial^2 z_k} = \sum_{i=1}^N w_{ik}^+ K_{ik} - \frac{\sum_{i=1}^N K_{ik}^2}{\sum_{i,j=1}^N K_{ij}}$.

We don't provide formulas for UMAP due to space limitation, but similarly to $t$-SNE, UMAP objective is also convex along $z_k$ with zero or one minimum depending on the sign of the second derivative at $z_k = 0$.

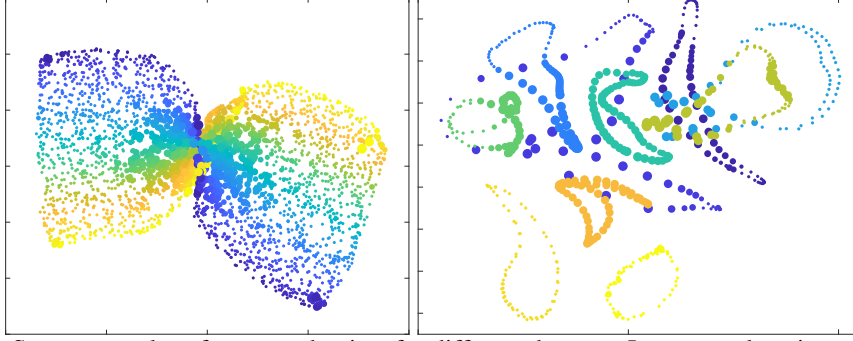

Figure 3: Some examples of pressured points for different datasets. Larger marker size corresponds to the higher pressure value. Color corresponds to the ground truth. *Left:* SNE embedding of the swissroll dataset with poor initialization that results in a twist in the middle of the roll. *Right:* 10 objects from COIL-20 dataset after 100 iteration of EE.

## 4 Pressured points for quality analysis

The analysis above can be directly applied to the existing algorithms as is, resulting in a qualitative statistic on the amount of pressure each point is experiencing during the optimization. A nice additional property is that computing pressured points can be done in constant time by reusing parts of the gradient. A practitioner can run the analysis for every iteration of the algorithm essentially for free to see how many points are pressured and whether the embedding results can be trusted.

In fig. 3 we show a couple of examples of embeddings with pressured points computed. The embedding of the swissroll on the left had a poor initialization that SNE was not able to recover from. Pressured points are concentrated around the twist in the embedding and in the corners, precisely where the difference with the ground truth occurs. On the right, we can see the embedding of the subset of COIL-20 dataset midway through optimization with EE. The embeddings of some objects overlap with each other, which results in high pressure.

In fig. 4 we show an embedding of the subset from MNIST after 200 iterations of $t$-SNE. We highlight some of the digits that ended up in clusters different from their ground truth. We put them in a red frame if a digit has a high pressure and in a green frame if their pressure is 0. For the most part the digits in red squares do not belong to the clusters where they are currently located, while digits in green squares look very similar to the digits around them.

## 5 Improving convergence by pressured points optimization

The analysis above can be also used for improvements in optimization. Imagine the embedding $\mathbf{X}$ has a set of points $\mathcal{P}$ that are pressured according to the definition above. Effectively it means that given a new dimension these points would utilize it in order to improve their current location. Let us create this new dimension $\mathbf{Z}$ with $z_k \neq 0$ for all $k \in \mathcal{P}$. Non-pressured points can only move along the original $d$ dimensions. For example, here is the augmented objective function for EE:

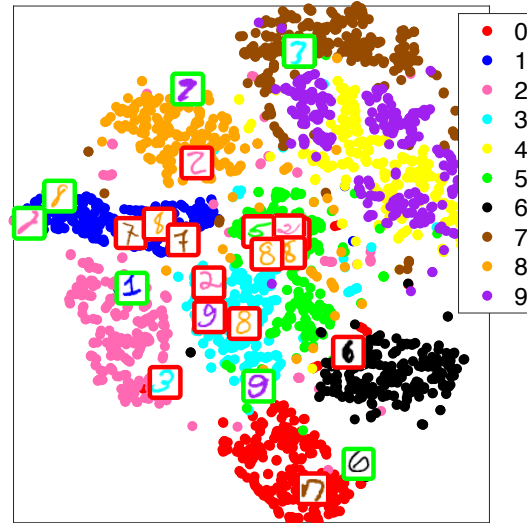

Figure 4: MNIST embedding after 200 iterations of $t$-SNE. We highlight two sets of digits located in clusters different from their ground truth: digits in red are pressured and look different from their neighbors; digits in green are non-pressured and look similar to their neighboring class.

$$\widetilde{E}(\mathbf{X}, \mathbf{Z}) = E(\mathbf{x}_{j\notin\mathcal{P}}) + E\left(\begin{pmatrix}\mathbf{x}_i\\\mathbf{z}_i\end{pmatrix}_{i\in\mathcal{P}}\right) + 2\Big(\sum_{i\in\mathcal{P}}\sum_{j\notin\mathcal{P}} w_{ij}^+ \|\mathbf{x}_i - \mathbf{x}_j\|^2$$
$$+ \lambda\sum_{i\in\mathcal{P}} e^{-z_i^2}\sum_{j\notin\mathcal{P}} w_{ij}^- e^{-\|\mathbf{x}_i-\mathbf{x}_j\|^2} + \sum_{i\in\mathcal{P}} z_i^2 \sum_{j\notin\mathcal{P}} w_{ij}^+\Big). \quad (7)$$

---

**Algorithm 1:** Pressured Points Optimization

---

**Input** : Initial $\mathbf{X}$, sequence of regularization steps $\boldsymbol{\mu}$.

Compute a set of pressured points $\mathcal{P}$ from $\mathbf{X}$ and initialize $\mathbf{Z}$ according to their pressure value.

**foreach** $\mu_i \in \boldsymbol{\mu}$ **do**

> **repeat**
>
> > Update $\mathbf{X}, \mathbf{Z}$ by minimizing $min\big(\widetilde{E}(\mathbf{X}, \mathbf{Z}) + \mu_i \mathbf{Z}\mathbf{Z}^T\big)$.
> > Update $\mathcal{P}$ using pressured points from new $\mathbf{X}$:
> > > 1. Add new points to $\mathcal{P}$ according to their pressure value.
> > > 2. Remove points that are not pressured anymore.
>
> **until** *convergence*;

**end**

**Output:** final $\mathbf{X}$

---

The first two terms represent the minimization of pressured and non-pressured points independently. The last term defines the interaction between pressure and non-pressured points and also has three terms. The first one gives the attraction between pressured and non-pressured points $\mathbf{X}$ in $d$ space. The second term captures the interactions between $\mathbf{Z}$ for pressured points and $\mathbf{X}$ for non-pressured ones. On one hand, it pushes $\mathbf{Z}$ away from $0$ as pressured and non-pressured points move closer to each other in $d$ space. On the other hand, it re-weights the repulsion between pressured and non-pressured points proportional to $\exp\left(-z_i^2\right)$ reducing the repulsion for larger values of $z_i$. In fact, since $\exp\left(-z^2\right) < 1$ for all $z > 0$, the repulsion between pressured and non-pressured points would always be weaker than the repulsion of non-pressured points between each other. Finally, the last term pulls each $z_i$ to $0$ with the weight proportional to the attraction between point $i$ and all the non-pressured points. Its form is identical to the $l_2$ norm applied to the extended dimension $\mathbf{Z}$ with the weight given by the attraction between point $i$ and all the non-pressured points.

Since our final objective is not to find the minimum of (7), but rather get a better embedding of $\mathbf{X}$, we are going to add a couple of additional steps to facilitate this. First, after each iteration of minimizing (7) we are going to update $\mathcal{P}$ by removing points that stopped being pressured and adding points that just became pressured. Second, we want pressured points to explore the new dimension only to the extent that it could eventually help lowering the original objective function. We want to restrict the use of the new dimension so it would be harder for the points to use it comparing to the original dimensions. It could be achieved by adding $l_2$ penalty to $\mathbf{Z}$ dimension as $\mu \sum_{i \in \mathcal{P}} z_i^2$. This is an organic extension since it has the same form as the last term in (7). For $\mu = 0$ the penalty is given as the weight between pressured and non-pressured points. This property gives an advantage to our algorithm comparing to the typical use of $l_2$ regularization, where a user has to resort to a trial and error in order to find a perfect $\mu$. In our case, the regularizer already exists in the objective and its weight sets a natural scale of $\mu$ values to try. Another advantage is that large $\mu$ values don't restrict the algorithm: all the points along $\mathbf{Z}$ just collapse to $0$ and the algorithm falls back to the original.

Practically, we propose to use a sequence of $\mu$ values starting at $0$ and increase proportionally to the magnitude of $d_k^+$, $k = 1 \ldots N$. In the experiments below, we set $step = 1/N \sum_k d_k^+$, although a more aggressive schedule of $step = \max(d_k^+)$ or more conservative $step = \min(d_k^+)$ could be used as well. We increase $\mu$ up until $z_k = 0$ for all the points. Typically, it occurs after 4–5 steps.

The resulting method is described in Algorithm 1. The algorithm can be embedded and run on top of the existing optimization methods for NLE: Spectral Direction and $N$-body methods.

In Spectral Direction the Hessian is approximated using the second derivative of $E^+$. The search direction has the form $\mathbf{P} = \left(4\mathbf{L}^+ + \epsilon\right)^{-1}\mathbf{G}$, where $\mathbf{G}$ is the gradient, $\mathbf{L}^+$ is the graph Laplacian defined on $\mathbf{W}^+$ and $\epsilon$ is a small constant that makes the inverse well defined (since graph Laplacian is psd). The modified objective that we propose has one more quadratic term $\mu\mathbf{Z}\mathbf{Z}^T$ and thus the Hessian for the pressured points along $\mathbf{Z}$ is regularized by $2\mu$. This is good for two reasons: it improves the direction of the Spectral Direction by adding new bits of Hessian, and it makes the Hessian approximation positive definite, thus avoiding the need to add any constant to it.

Large-scale $N$-Body approximations using Barnes-Hut [Yang et al., 2013, van der Maaten, 2014] or Fast Multipole Methods (FMM, Vladymyrov and Carreira-Perpiñán, 2014) to decrease the cost of objective function and the gradient from $\mathcal{O}(N^2)$ to $\mathcal{O}(N \log N)$ or $\mathcal{O}(N)$ by approximating the interaction between distant points. Pressured points computation uses the same quantities as the gradient, so whichever approximation is applied carries over to pressured points as well.

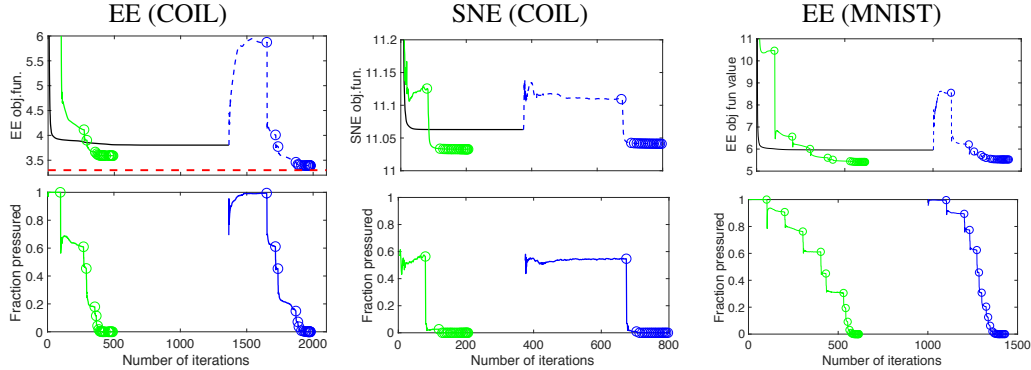

Figure 5: The optimization of COIL-20 using EE (*left*) and SNE (*center*), and optimization of MNIST using EE (*right*). Black line shows the SD, green line shows PP initialized at random, blue line shows PP initialized from the local minima of the SD. Dashed red line indicates the absolute best result that we were able to get with homotopy optimization. Top plots show the change in the objective function, while the bottom show the fraction of the pressured points for a given iteration. Markers 'o' indicate change of $\mu$ value.

## 6 Experiments

Here we are going to compare the original optimization algorithm, which we call simply spectral direction (SD)[2] to the Pressured Point (PP) framework defined above using EE and SNE algorithms. While the proposed methodology could also be applied to $t$-SNE and UMAP, in practice we were not able to find it useful. $t$-SNE and UMAP are defined on kernels that have much longer tails than the Gaussian kernel used in EE and SNE. Because of that, the repulsion between points is much stronger and points are spread far away from each other. The extra-space given by new dimension is not utilized well and the objective function decrease is similar with and without the PP modification.

For the first experiment, we run the algorithm on 10 objects from COIL-20 dataset. We run both SNE and EE 10 different times with the original algorithm until the objective function does not change for more than $10^{-5}$ per iteration. We then run PP optimization with two different initializations: same as the original algorithm and initialized from the convergence value of SD. Over 10 runs for EE, SD got to an average objective function value of $3.84 \pm 0.18$, whereas PP with random initialization got to $3.6 \pm 0.14$. Initializing from the convergence of SD, 10 out of 10 times PP was able to find better local minima with the average objective function value of $3.61 \pm 0.19$. We got similar results for SNE: average objective function value for SD is $11.07 \pm 0.03$, which PP improved to $11.03 \pm 0.02$ for random initialization and to $11.05 \pm 0.03$ for initialization from local minima of SD.

In fig. 5 we show the results for one of the runs for EE and SNE for COIL. Notice that for initial small $\mu$ values the algorithm extensively uses and explores the extra dimension, which one can see from the increase in the original objective function values as well as from the large fraction of the pressured points. However, for larger $\mu$ the number of pressured points drops sharply, eventually going to $0$. Once $\mu$ gets large enough so that extra dimension is not used, optimization for every new $\mu$ goes very fast, since essentially nothing is changing.

As another comparison point, we evaluate how much headroom we can get on top improvements demonstrated by PP algorithm. For that, we run EE on COIL dataset with homotopy method [Carreira-Perpiñán, 2010] where we performed a series of optimizations from a very small $\lambda$, where the objective function has a single global minimum, to final $\lambda = 200$, each time initializing from the previous solution. We got the final value of the objective function around $E = 3.28$ (dashed red line on the EE objective function plot on fig. 5). While we could not get to a same value with PP, we got very close with $E = 3.3$ (comparing to $E = 3.68$ for the best SD optimization).

Finally, on the right plot of fig. 5 we show the minimization of MNIST using FMM approximation with $p = 5$ accuracy (i.e. truncating the Hermite functions to $5$ terms). PP optimization improved the convergence both in case of random initialization and for initialization from the solution of SD. Thus, the benefits of PP algorithm can be increased by also applying SD to improve the optimization direction and FMM to speed up the objective function and gradient computation.

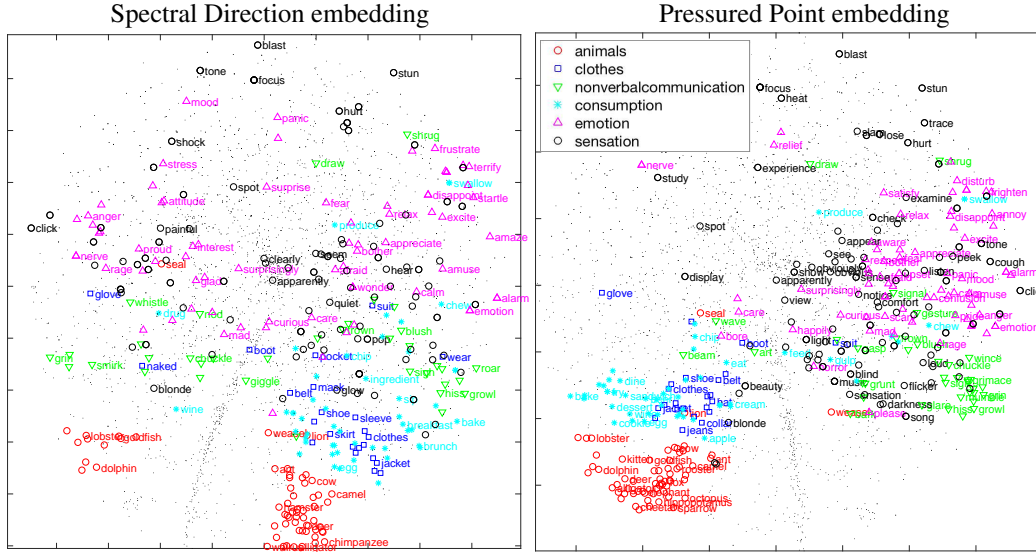

Figure 6: Embedding of the subset of word2vec data using EE optimized with SD and further refined by PP. We highlight six word categories that were affected the most by embedding adjustment.

As a final experiment, we run the EE for word embedding vectors pretrained using word2vec [Mikolov et al., 2013] on Google News dataset. The dataset consists of $200\,000$ word-vectors that were downsampled to $5\,000$ most popular English words. We first run SD 100 times with different initialization until the embedding does not change by more than $10^{-5}$. We then run PP, initialized from SD. Fig. 6 shows the embedding of one of the worst results that we got from SD and the way the embedding improved by running PP algorithm. We specifically highlight six different word categories for which the embedding improved significantly. Notice that the words from the same category got closer to each other and formed tighter clusters. Note that more feelings-oriented categories, such as *emotion*, *sensation* and *nonverbalcommunication* got grouped together and now occupy the right side of the embedding instead of being spread across. In fig. 7 we show the final objective function values for all 100 runs together with the

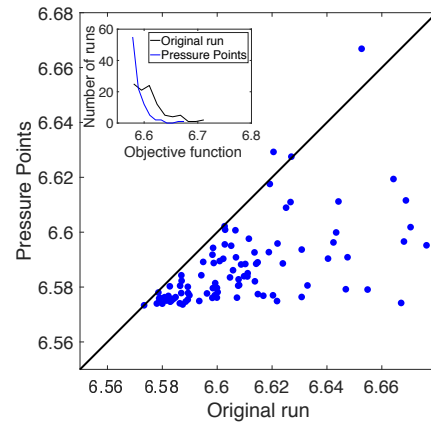

Figure 7: The difference in final objective function values between PP and SD for 100 runs of `word2vec` dataset using EE algorithm. See main text for description.

improvements achieved by continuing the optimization using PP. In the inset, we show the histogram of the final objective function values of SD and PP. While the very best results of SD have not improved a lot (suggesting that the near-global minimum has been achieved), most of the times SD gets stuck in the higher regions of the objective function that are improved by the PP algorithm.

## 7 Conclusions

We proposed a novel framework for assessing the quality of most popular manifold learning methods using intuitive, natural and computationally cheap way to measure the pressure that each point is experiencing from its neighbors. The pressure is defined as a minimum of objective function when evaluated along a new extra dimension. We then outlined a method to make use of that extra dimension in order to find a better embedding location for the pressured points. Our proposed algorithm is able to get to a better solution from a converged local minimum of the existent optimizer as well as when initialed randomly. An interesting future direction is to extend the analysis beyond one extra dimension and see if there is a connection to the intrinsic dimensionality of the manifold.

### Acknowledgments

I would like to thank Nataliya Polyakovska for initial analysis and Makoto Yamada for useful suggestions that helped improve this work significantly.

## Footnotes

[1]Instead of the classic SNE, in this paper we are going to use symmetric SNE [Cook et al., 2007], where each probability is normalized by the interaction between all pairs of points and not every point individually.

[2]It would be more fair to call our method SD+PP, since we also apply spectral direction to minimize the extended objective function, but we are going to call it simply PP to avoid extra clutter.

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
