[Reviews · NeurIPS 2019]

Reviewer 1



-- Post author-feedback comments -- I appreciate the authors taking the time to address the optional improvements I suggested. I think the addition of UMAP will improve the paper, and I would like to the encourage the authors to keep exploring the other discussed aspects in future work. In my initial review I already strongly recommended to accept this paper, as I still do now, so my overall score remains unchanged. -- Initial review -- This work aims to alleviate the sensitivity of popular dimensionality reduction approaches such as SNE and tSNE to initialization and local minima in the optimization of low dimensional coordinates to fit the neighborhood structure of the original (high dimensional) data. In particular, the authors focus on identifying points that get stuck in suboptimal positions due to the inability of the optimization to find a low dimensional path that will ultimately improve the placement of the points, even though in the short term it might incur higher penalty due to the way attractive and repulsive forces interact in the optimization loss (hence the local minima). To address such cases, the authors define a notion of pressure applied to data points by the dimensionality of the embedded coordinates, and propose to release such pressure by temporarily allowing the optimization process to use an additional auxiliary dimension for identified pressured points. Then, as the optimization progresses, a regularization is applied to gradually restrict the use of such auxiliary dimensions. The manuscript is well written, well motivated, and convincingly establishes the reasoning behind the proposed approach as well as its effectiveness. Further, the problem dealt with here is indeed timely and crucial for the reliability of visualization methods based on SNE, tSNE, and other variations. Therefore, I recommend this paper be accepted and look forward to seeing it presented at NeurIPS. Optional improvements appear in a separate part of the review.

Reviewer 2



The paper is written rather clearly, and I think the method itself is interesting from a practical point of view, although it will be great if there is more theory to it.

Reviewer 3



I have read the authors' response and other reviewers' comments. I choose to raise my score. ------------------------------------------------------------- Pros: - The idea of identifying pressure points is new in the context of nonlinear dimension reduction. The way these points are utilized is also inspirational. - Extensive results are run on several datasets. Cons: - Like many other work in nonlinear dimension reduction, this paper does not provide a systematic, convincing way to evaluate and compare the new approach against previous methods which are claimed to be less effective. The visual comparison in Figure 6 appears cherry-picking in methodology. In the second comparison experiments, even with this dimension increasing strategy, the algorithm failed to identify the unique global minimum, which is frustrating; the authors claimed the objective function value was close to optimal but it makes little sense without interpretable units. - It may well be the case that the premise of embedding all high-dimensional points into a uniform low-dimensional space is misleading. As demonstrated in the motivation and experiments in this paper, if local minimum is really a concern for nonlinear dimension reduction algorithms, maybe one should use different dimensions for different points. - Are there interpretable properties associated with those "pressure points"? Is there any particular explanation for why they become stubborn during the optimization procedure? Do they depend on initialization and randomness of the algorithm? Only with such in-depth questions (at least partially) addressed will this work be justified as a solid contribution instead of a coincidental case study. - Before applied to real data, algorithms should be first applied to synthetic data for which one knows what the "ground truth" to expect. An algorithm that successfully recovers the expected truth can then be trusted. Otherwise, it is difficult to argue the algorithm works in real data, as for such high-dimensional datasets it is not unlikely that some correlations will create non-existent patterns captured by human perception.

[Author Response · NeurIPS 2019]



Figure 1: Pressured points for different datasets. For each of four datasets, left plot is the ground truth and the right plot is an embedding with marker size indicating pressure value. Color corresponds to the ground truth.

We thank the reviewers for the time. We are really glad that the reviewers have found that the paper provides a novel
idea, is timely, is well written and motivated, and has extensive results.

**Reviewer #1** *Results for UMAP*. Indeed, the objective function of UMAP is similar to $t$-SNE and can be written as

$$E_{\text{UMAP}}(\mathbf{X}) = \sum_{i,j}(\log(1 + a\left\|\mathbf{x}_i - \mathbf{x}_j\right\|^{2b})) + \sum_{i,j}\log(1 - (1 + a\left\|\mathbf{x}_i - \mathbf{x}_j\right\|^{2b})^{-1}). \tag{1}$$

Similarly to other methods, this function also has the property of a single global minimum along a new dimension $\mathbf{Z}$ that
could be found with a few iterations of Newton's method. We will make sure to update the paper with this information.

*Robustness.* We certainly hope that our approach would reduce the amount of "tsne engineering". We were hesitant
to include any claims of robustness, since after all we are dealing with highly non-convex obj. fun. with many local
minima. One would only hope to find a global solution. Our intuition does suggest that *all* of the local minima are
produced by some points being pressured, however we were not yet able to prove it. In fig. 7 of the main paper, one can
see that the variance of the final obj. fun. values of PP is smaller than the one from SD, however it is not exactly zero.

*More aux dimensions.* Mathematically, nothing prevents us from computing pressure points recursively one after
another, up until all the points become non-pressured. Practically however, we would have to optimize the embedding
separately for each dimension, which is costly. Our goal was to create a practical algorithm that could improve the
results of existing methods, thus we have settled on increasing the dimensionality only by one.

**Reviewer #4** *Comparison to other methods.* We do not propose a novel dimensionality reduction technique, but
rather give insights and offer a novel optimization to the *existing* methods. Thus, the baseline should be given by the
state-of-the-art optimization method (Spectral Direction), comparison to which we provide.

*Results for Figure 6.* We highlighted categories that differ the most from one embedding to another according to the
Procrustes alignment error. The embedding for all these categories got improved (theoretically they could have gotten
worse, but they did not). This is an important point and we will clarity it better in the paper.

*Global minimum.* The obj. fun. of the embedding methods is highly non-convex and finding a global minimum exactly
is a very hard problem (see note on *Robustness* above). In fig. 5 we show the best possible results that we were able to
get with a very careful and slow optimization. PP was able to get to a similar solution much faster.

*Using higher-dimensional embeddings.* The number of dimensions are often given as a hard constraint by the user. For
example, one of the most typical application for the dimensionality reduction methods is the data visualization where
the embedding dimensionality has to be equal two or three. For these cases, the goal is to find (potentially very lossy)
embedding that would best represent the structure of the data. Finding the latent dimensionality is out of the scope of
this paper (see also *More aux dimensions* above).

*Interpretability.* We discussed a typical scenario of the way pressured points arise in fig. 2 of the main paper and in
the beginning of section 3. In addition, in fig. 3 we provided some examples of the pressure points for some synthetic
dataset. As per reviewer suggestion, in fig. 1 above we include some additional examples of pressure points on synthetic
data. Notice that the points become pressured when they are far from ground truth and are located "on top" of other
points. In all the cases shown (except for the swiss roll with a hole), the original method (SNE) got stuck in a local
minima. Our method was able to get out of it and achieve results that are almost identical to the ground truth.

[Meta-Review · NeurIPS 2019]

Dimensionality reduction such as tSNE is widely used to visualize and interpret (and often over interpret) high-dimensional data. Thus such visualization has become a staple in the field and it is has been a while since I have seen substantial progress in improving such visualization techniques and this paper is such a case. Reviewer 1 summarizes the contribution and its importance better than I could word it myself: This work has two main contributions, which are sufficiently significant given the interest in visualization and dimensionality reduction via SNE, tSNE, and further extensions: 1. Identification of pressure points that are "stuck" in suboptimal location in the embedding due to local minima caused by dimensionality constraints. 2. Improved optimization process for SNE, tSNE, and similar methods, by allowing pressure points to temporarily bypass the dimensionality constraints in order to alleviate local minima and provide a more robust embedding. The manuscript is well written, well motivated, and convincingly establishes the reasoning behind the proposed approach as well as its effectiveness. All three reviewers agree on accepting the paper. All reviewers agreed that the paper provided new insights, a novel approach, a valuable practical contribution which is extensively validated on multiple datasets and is well written.